# Chemical and Morphological Composition of Norway Spruce Wood (*Picea abies*, L.) in the Dependence of Its Storage

**DOI:** 10.3390/polym13101619

**Published:** 2021-05-17

**Authors:** Iveta Čabalová, Michal Bélik, Viera Kučerová, Tereza Jurczyková

**Affiliations:** 1Department of Chemistry and Chemical Technologies, Faculty of Wood Sciences and Technology, Technical University in Zvolen, T. G. Masaryka 24, 960 53 Zvolen, Slovakia; xbelik@is.tuzvo.sk (M.B.); viera.kucerova@tuzvo.sk (V.K.); 2Department of Wood Processing, Czech University of Life Sciences in Prague, Kamýcká 1176, 16521 Praha, Czech Republic; jurczykova@fld.czu.cz

**Keywords:** spruce wood, cellulose, hemicelluloses, lignin, extractives, time of storage, fiber characteristics

## Abstract

Chemical composition and morphological properties of Norway spruce wood and bark were evaluated. The extractives, cellulose, hemicelluloses, and lignin contents were determined by wet chemistry methods. The dimensional characteristics of the fibers (length and width) were measured by Fiber Tester. The results of the chemical analysis of wood and bark show the differences between the trunk and top part, as well as in the different heights of the trunk and in the cross section of the trunk. The biggest changes were noticed between bark trunk and bark top. The bark top contains 10% more of extractives and 9.5% less of lignin. Fiber length and width depends on the part of the tree, while the average of these properties are larger depending on height. Both wood and bark from the trunk contains a higher content of fines (fibers <0.3 mm) and less content of longer fibers (>0.5 mm) compared to the top. During storage, it reached a decrease of extractives mainly in bark. Wood from the trunk retained very good durability in terms of chemical composition during the storage. In view of the morphological characteristics, it occurred to decrease both average fibers length and width in wood and bark.

## 1. Introduction

Wood is an anisotropic material, with respect to its anatomical, physical, and chemical properties, and is made up of different kinds of cells. Wood is degradable by fungi, microorganisms and heating [1]. Degradation of wood and its chemical structure are influenced by the temperature, oxygen available to the material, ambient pressure, wood type and shape, moisture content of the wood, and additives, such as inorganic substances, sorbed emissions, etc. [2,3,4]. Fedyukov et al. [5] progressively describe the decreasing of cellulose content of spruce wood with soil condition deterioration.

The structure of wood, chemical components (cellulose, hemicelulloses, lignin and extractives) and their relative mass proportion depending on the morphological region, kind of the tree, and age of the wood [1]. Variation can be found within a single tree from the center of the trunk to the bark, from the trunk to the top, between earlywood and latewood, and between sapwood and heartwood [6]. According to Fengel and Wegener [7], earlywood contains more lignin and inversely less cellulose than latewood. 

Understanding the morphological and chemical heterogeneity of wood is important in its utilization, for example in the paper industry. Like wood, bark is an important source of raw material and chemical compounds [8]. The chemical composition of bark varies among the different tree species and depends on the morphological element involved. Many of the constituents present in wood also occur in bark, but their proportion is different. Bark can roughly be divided into the fraction: fibers, corok cells and fine substance (including parenchyma cells) [1]. Bark contains much more extractives than wood from the trunk [9], considerable amounts of bioactive components such as antioxidants (polyphenols), and also structural polysaccharides such as pectins [10,11,12]. Degradation of biomass is influenced by photodegradation (UV light), high temperatures during storage in piles, microorganizms, etc., Routa et al. [13] describe the major reactions of lipophilic extractive compounds during wood storage, which can be divided into three types: (1) hydrolysis of triglycerides (rapid reaction) and steryl esters and waxes (proceeds slower); (2) oxidation/degradation/polymerization of resin acids, unsaturated fatty acids, and to some extent, other unsaturated compounds; and (3) evaporation of volatile terpenoids, mainly monoterpenes.

As any wood material, spruce wood is chemically complex and its physical, chemical and morphological characteristics are not uniform. In Slovakia, it is the most common wood specie. In Norway spruce, approximately 95% of wood cell matrix is composed of tracheids, which can also be termed fibers. The average tracheid length is mainly influenced by tree age. As most wood properties, fibers length and width varies greatly both within and among trees, depending on its vertical and radial position (ring age) in trunk, and forest stand [14]. The tracheids length is shortest next to the pith, the increase with age is at first very rapid in the juvenile period, then slow down between the ages 10–30 and thereafter, as mature wood begins to form, increases very gradually with seasonal fluctuation [15].

The aim of this paper is to characterize chemical composition (extractives, lignin, polysaccharides, cellulose and hemicelluloses) and morphological properties (fibers length and width) of different parts of Norway spruce wood (*Picea abies* (L.) Karst.) and to evaluate the rate of changes in these properties after wood storage. 

## 2. Experimental

### 2.1. Materials

Norway spruce wood (*Picea abies* L. Karst.) was harvested in the middle part of Slovakia (Zvolen region) in June. The 65-year-old tree was cut 0.5 m from the ground and its height was 25 m. For the experiment, we used a 5 m long trunk part (the diameter of 32 cm—thicker part and the diameter of 29.5 cm—thinner part) and the top part of tree, represented the waste biomass (the diameter of 12 cm—thicker part and the diameter of 7.5 cm—thinner part). We took samples (cut with an all diameter) from the trunk in a height of 0.5, 1.5, 2.5, 3.5 and 4.5 m from the ground and both wood and bark from the top. Equally, we took samples A, B, C (Figure 1) in a height of 0.5 and 4.5 m, whereas sample A represent juvenile wood, which is within the first 20 years of growth; B sample, the next 20 years; and C sample, the rest. 

After trunk sampling, we obtained five wood pieces with a length of approximately 1 m. We debarked the first one. A part of the bark was used for the experiment and the rest for storage. The rest (four) of the wood pieces (in the bark) were also used for storage. After 2, 4, 6 and 8 months of storage, we always analysed one piece of wood (the first sample was wood and the next sample bark from this piece) and bark, which was stored separately. Specifications of the samples are described in Table 1.

During storage, the conditions were as follows (Table 2):

### 2.2. Methods

#### 2.2.1. Chemical Composition of Wood

The samples were disintegrated into sawdust, and fractions 0.5 mm to 1.0 mm in size were used for the chemical analyses. The extractives content (EL) was determined in a Soxhlet apparatus with a mixture of ethanol and toluene (2:1) according to ASTM D1107-96 [16]. The lignin content (LIG) was determined according to Sluiter et al. [17], and the cellulose content (CEL) was determined according to the method by Seifert [18], and the holocellulose content according to the method by Wise et al. [19]. Hemicelluloses (HEMI) were calculated as the difference between the holocellulose and cellulose contents. Measurements were performed on four replicates per sample. The results were presented as oven-dry wood percentages.

#### 2.2.2. Fibers Length and Width

Two hundred-millilitre mixtures of concentrated CH_3_COOH and 30% H_2_O_2_ (1:1, *v*/*v*) were poured onto the wood samples (weight = 10 g and dimensions = 20 mm × 2 mm × 2 mm). Then, the samples were refluxed for 3 h, suction filtered through a sintered glass filter (S1), and washed with distilled water. An L & W Fiber Tester (Lorenzen and Wettre, Kista, Sweden) was used to determine the fibre dimensional characteristics. This measurement is based on the principle of two-dimensional imaging technology. The measurement technology is automated, allowing for frequent and rapid analysis of the fibre quality. The instrument measures various fibre properties, such as the length and width of the fibres, fine portion (from 0.1 mm to 0.2 mm). Measurements were performed on a single replicate per sample, and the number of fibres within each population of the replicate ranged from 19,182 cells to 21,128 cells.

## 3. Results and Discussion

### 3.1. The Differences in the Spruce Wood and Bark in the View of Its Chemical Composition and a both Fibers Length and Width Distribution

Wood is composed of cellulose, hemicelluloses, lignin, and extractives. From the results of the chemical analysis of spruce wood (Figure 2 and Figure 3), we obtained the differences between the wood, as part of the trunk, and the top part of the tree. We compared the chemical composition of the trunk part in a height of 0.5 m and 4.5 m. We have taken samples (A, B, C) from the tree trunk, as shown Figure 1. The results indicate that there are differences in extractives content between A, B, and C in a height of 0.5 m. The biggest content of extractives, 1.68%, is in the part C (the last 25 years of growth), then A, 1.29%, and the last B, 1.04%. In a height of 4.5 m, there are no differences between the A and B part, and the amount of extractives in part C is comparable to the results in a height of 0.5 m. The amount of LIG, CEL and HEMI was very similar in a height of 0.5 and 4.5 m as well (Figure 2).

The analysis of chemical composition of wood and bark in different heights (Figure 3) exposed that the extractive content of both wood and bark is growing up with a height of the tree. The biggest extractives content of 32.35% was recorded in bark taken from the top part. Several authors mention the extractives amount of spruce bark from 23.5% to 28.3%, depending on the part of bark (inner bark from 17.3 to 38.7%, outer from 19.1 to 43.3%) [20,21,22,23]. The extractives content of the wood part is between 1% and 4.5% [24,25], while there is a difference between sapwood, values from 1.7% to 2.7% and heartwood, from 1.1% to 1.8% [26].

The amount of main chemical compounds: LIG (from 23.69 to 26.06%), CEL (from 39.01 to 42.51%), and HEMI (from 34.98 to 35.30%) were very similar in wood samples taken from the trunk (in a height from 0.5 to 4.5 m) compared to the top part. Sjostrom [27] mentions the lignin content of 27.5%, cellulose of 39.5 and hemicelluloses of 17.2% for Picea glauca; Wang et al. [28] showed a Klasson lignin of 28.2% for *Picea abies* (L.) H. Karst; Neiva et al. [29] showed a lignin content of 27.22%, polysaccharides of 71.18% for *Picea abies* (L.) H. Karst; Harris [30] showed a lignin from 28.0% to 30.8%, cellulose content from 38.1% to 40.3% for juvenile wood, and the lignin from 26.1% to 28.2%, cellulose from 40.2% to 42.7% for the mature wood of Norway spruce.

The biggest changes in chemical composition we noticed between bark trunk and bark top (Figure 3). As mentioned above, the extractive content was higher in the bark-top (32.35%) compared to the bark-trunk (22.16%), the lignin content was higher in bark-trunk (24.97%) compared to bark-top (15.46%), and the content of polysaccharides were similar (52.87% bark-trunk and 52.18% bark-top). Neiva et al. [29] determined the lignin between 26.86% and 29.92%, and polysaccharides content between 37.86% and 52.27%, depending on the bark fraction.

Results of the Fiber Tester analysis (Figure 4 and Figure 5) show that there are differences in both wood fibres length and width, mainly between samples A and B in a height of 0.5 and 4.5 m. The lower part of the trunk contains the highest amount of fines (>0.3 mm), from 45.94% (part B) to 52.38% (part A). The lowest content of long fibers (more than 1.01 mm) we determined in sample 0.5-A. There are also visible differences of fibers width in this sample (Figure 5). This part of the wood contains a higher amount of narrow fibers compared to sample 4.5-A. The average fiber length depends on the part of the trunk. The results show (Table 3) the smallest average fiber length and width in sample A, then B, and lastly C in a height of 0.5 m and 4.5 m as well. Both average fiber length and width are larger with the height of the tree.

Wood from the trunk contains the highest amount of fines, approximately 50% (Figure 4) and its average length is 1.5 mm (Table 4). Tyrväinen [31] presents the average fiber length and its variation of Norway spruce trunk, namely inner heartwood 1.9 mm (1.28–2.70 mm), middle zone 3.0 mm (1.69–3.88 mm), and outer sapwood 3.7 mm (2.80–4.29 mm). Harris [30] and Lönnberg et al. [32] mentioned the fiber width of Norway spruce as being between 15.0 and 28.5 µm in juvenile wood, and between 29.3 and 39.7 µm in mature wood. According to our results (Figure 6), the biggest content of fraction (from 45% wood-trunk to 60.85% wood-top) is in the width class from 15.1 to 30.0 µm and the average width of the wood trunk is 24.33 µm and bark 21.02 µm (Table 4).

The results of the bark analysis (Figure 6 and Figure 7) show the differences mainly in fibre length distribution. The results of fibers width distribution were very similar and the largest proportion of fibers is in the class from 15.1 to 30 µm (67.53–69.17%). Bark from the trunk contains a high amount of fines (65.9%) and a lower amount of longer fibers compared to the bark from the top part of the tree. Bark contains a higher amount of shorter fibers to 0.5 mm (65.88% bark-top, 80.81% bark-trunk), than the wood part.

### 3.2. The Differences in the Spruce Wood and Bark after Storage

The amount of extractives is influenced by wood storage. Tree bark is a biological, heterogeneous material whose composition is changing. Immediately after tree harvesting, the amount of volatiles declined, and degradation continued during wood storage.

According to the results (Figure 8 and Figure 9), it is visible that the biggest changes in the chemical composition can be obtained in bark (stored separately and as a part of the trunk). Figure 10 explains the results of chemical composition of wood trunk, while values are very comparable. Spruce wood retained very good durability in terms of its chemical composition after eight months of storage. Several authors studied wood with different natural aging time and the results found that the proportion of saccharides gradually decreases (mainly due to the hemicelluloses degradation), and the content of lignin increases successively with increasing time [33,34,35].

The degradation of main chemical components of both bark stored separately and bark as a part of the trunk, was obvious. During eight months of storage, a decreasing of extractives occurred (bark of 80.96%, bark trunk of 73.69%) and HEMI (bark of 67.52%, bark trunk of 49%), and increasing of LIG (bark of 45.65%, bark trunk of 60.9%) and CEL (bark of 65.73% and bark trunk of 69.11%). Bergström and Matison [36] in their study describe a decrease in the extractive content during storage, roughly halving during the first four weeks. The biggest losses noticed during this period were in the amounts of hydrophilic, and lipophilic as well. From chemical components, the stilbenes are very sensitive to degradation [37]. According to Routa et al. [13], after eight weeks of pine bark storage, we obseved types of extractive substances, which are predominate in the bark: triglycerides, steryl esters, sterols, resin acids, and fatty acids.

The Fiber Tester analysis (Figure 11 and Figure 12) shows changes in fibers length and width distribution of both wood and bark. Sample of wood stored for eight month contains lower amount of fibers longer than 1.01 mm (decrease of 23,9%). The both average fiber length and width of samples decreased during storage (Table 4) because of wood and bark degradation.

## 4. Conclusions

The amount of extractives is the highest in the last 25 years of growth (close to bark), polysaccharides and lignin content was similar in a cross section of trunk.Compared to the chemical composition of wood trunk in a different height (0.5, 1.5, 2.5, 3.5 and 4.5 m from the ground), very similar results of cellulose, hemicelluloses and lignin content were determined. The extractive content of both wood and bark is growing up with the height of the tree.From the view of chemical composition, the differences between bark, as a part of the trunk and the top part of the tree were obtained. In the bark top we determined bigger amount of extractives (10%) and less amount of lignin (9.5%) compared to the bark trunk. The amount of polysaccharides was similar.Juvenile wood contains a smaller amount of longer fibers, >1.01 mm, than other parts of the tree.The both average fiber length and width is higher with a height of the tree.The amount of extractives is very influenced by the time of storage, especially in bark. During eight months of storage, a decreasing of extractives occurred (from 73.7 to 80.9%) and hemicelluloses (from 49.1 to 67.5%) relative content, and an increasing of lignin and cellulose. Bark stored separately degraded faster than bark stored on the trunk.In view of the chemical composition of the wood from the trunk retained very good durability during the storage for eight months.Eight months stored wood contains a lower amount of fibers, longer than 1.01 mm compared to raw wood. The both average fibers length and width decreased during storage.

## Figures and Tables

**Figure 1 polymers-13-01619-f001:**
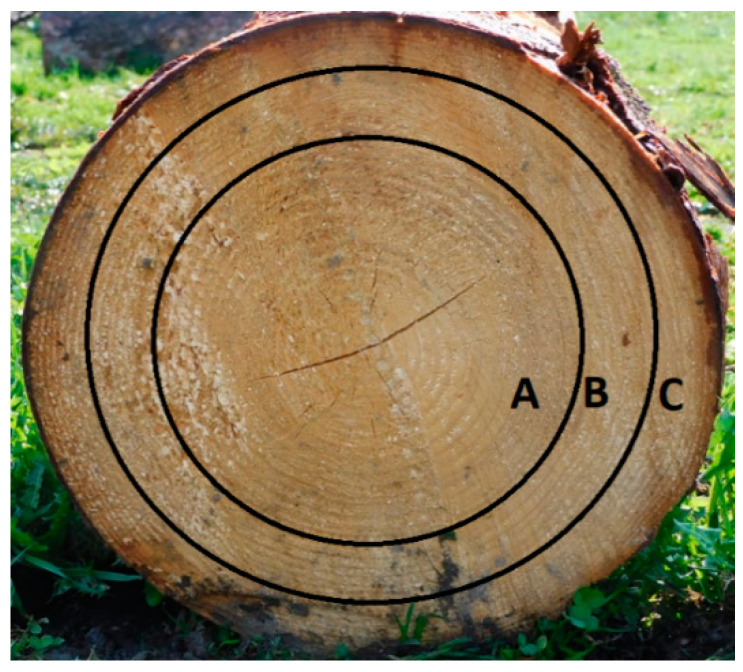
Samples A, B, and C used for analysis.

**Figure 2 polymers-13-01619-f002:**
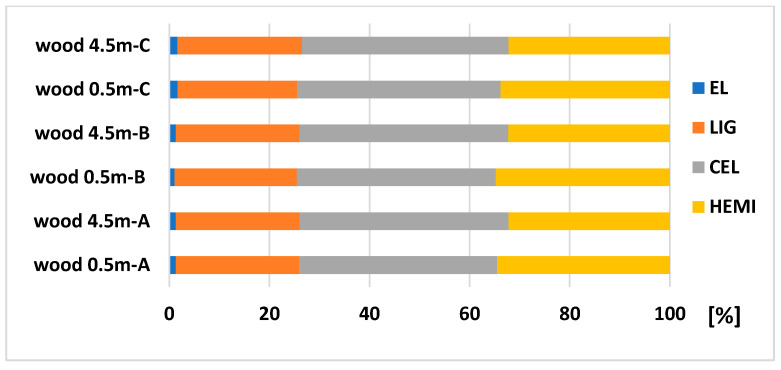
Chemical composition (relative values) of spruce wood trunk at heights of 0.5 and 4.5 m.

**Figure 3 polymers-13-01619-f003:**
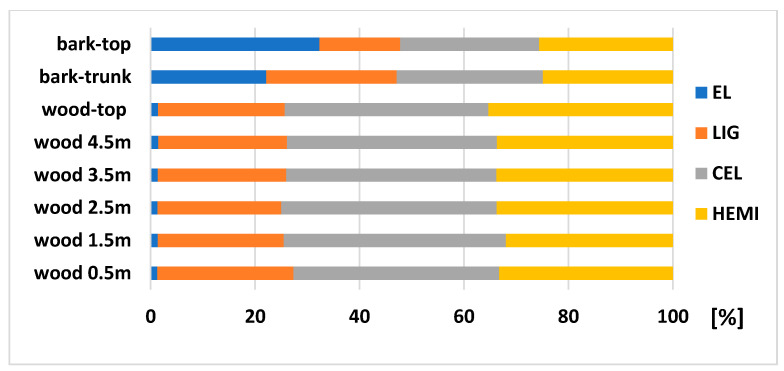
Chemical composition (relative values) of spruce wood and bark at different heights.

**Figure 4 polymers-13-01619-f004:**
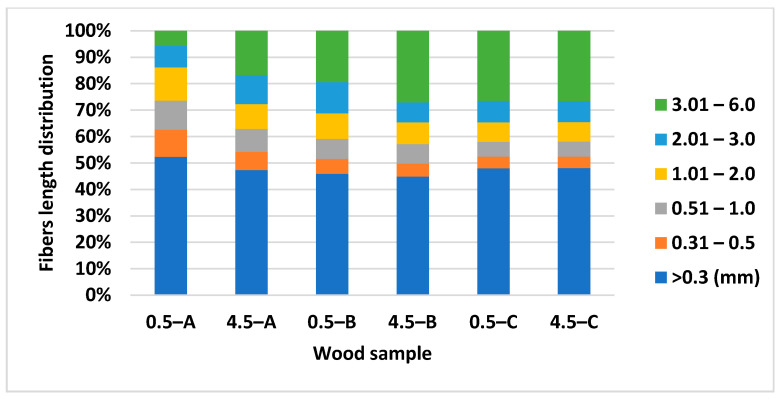
Fibers length distribution of spruce wood trunk at heights of 0.5 and 4.5 m. Fibres length classes: >0.3; 0.31–0.5; 0.51–1.0; 1.01–2.0; 2.01–3.0; 3.01–6 mm.

**Figure 5 polymers-13-01619-f005:**
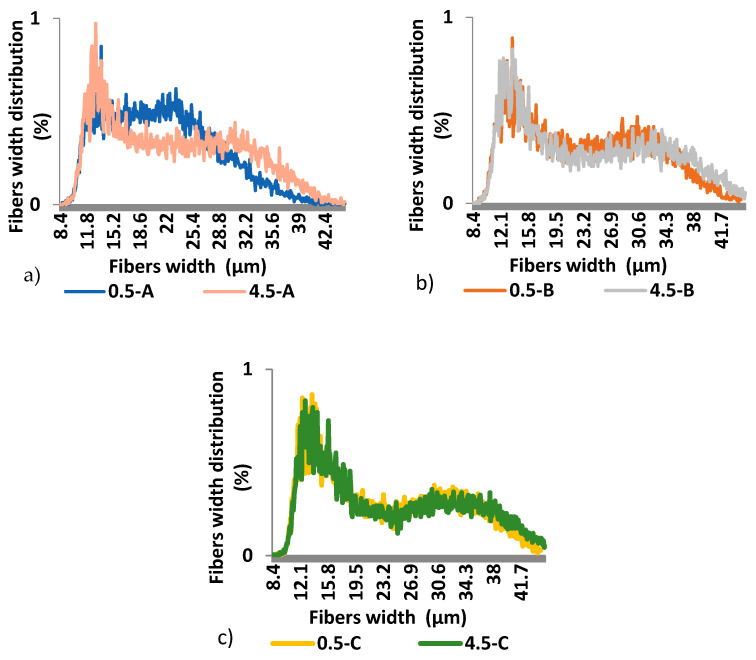
Fibers width distribution of spruce wood trunk at heights of 0.5 and 4.5 m; (**a**) samples 0.5-A, 4.5-A; (**b**) samples 0.5-B, 4.5-B; (**c**) samples 0.5-C, 4.5-C.

**Figure 6 polymers-13-01619-f006:**
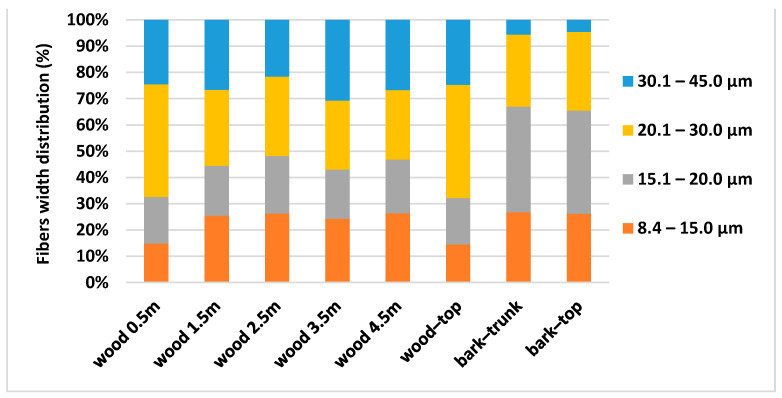
Fibres width distribution of spruce wood and bark at different heights. Fibres width classes: 8.4–15.0 µm; 15.1–20.0 µm; 20.1–30.0 µm; 30.1–45.0 µm.

**Figure 7 polymers-13-01619-f007:**
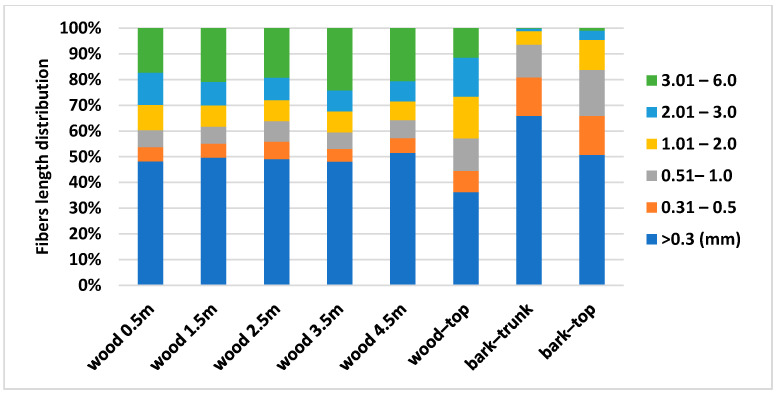
Fibres length distribution of spruce wood and bark at different heights.

**Figure 8 polymers-13-01619-f008:**
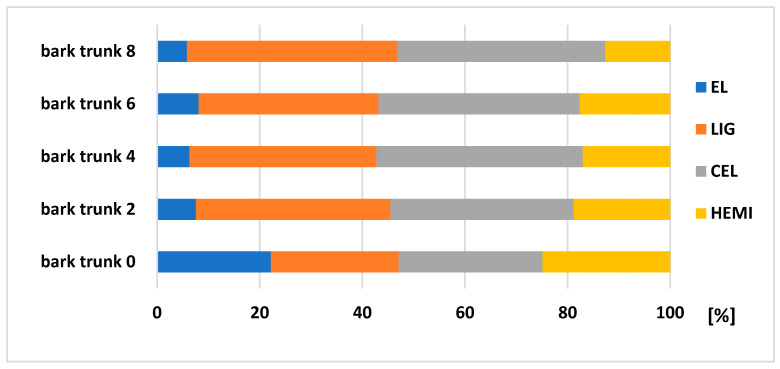
Changes in the composition (relative values) of spruce bark from trunk after its storage of 2, 4, 6, and 8 months.

**Figure 9 polymers-13-01619-f009:**
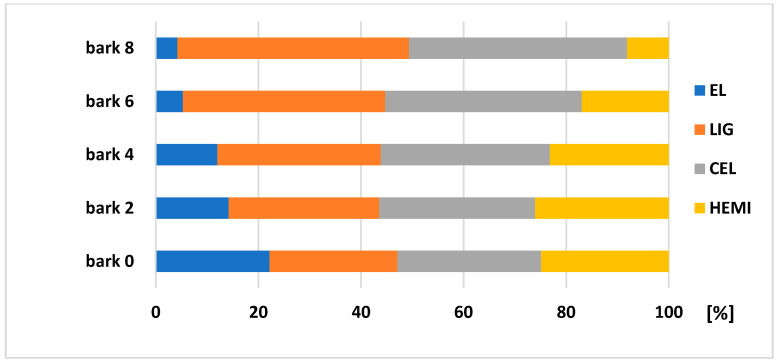
Changes in the composition (relative values) of spruce bark after its storage of 2, 4, 6, and 8 months.

**Figure 10 polymers-13-01619-f010:**
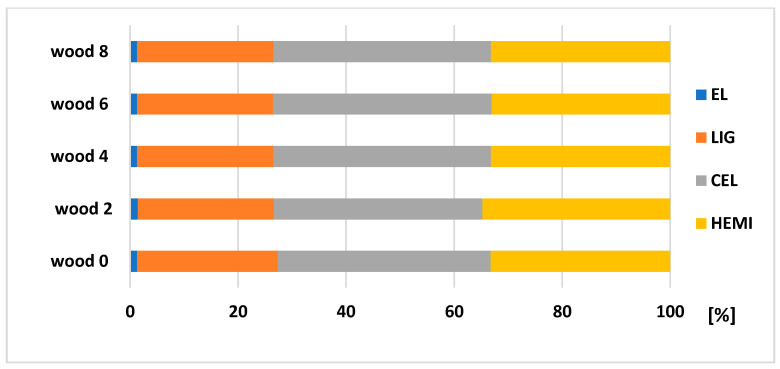
Changes in the spruce wood composition (relative values) after its storage of 2, 4, 6, and 8 months.

**Figure 11 polymers-13-01619-f011:**
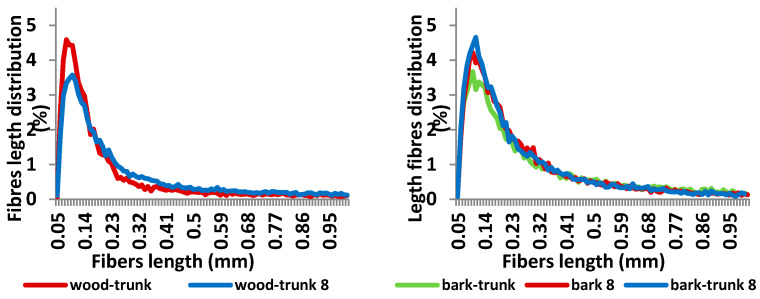
Fibres length distribution of spruce wood trunk and bark before and after its storage of 8 months.

**Figure 12 polymers-13-01619-f012:**
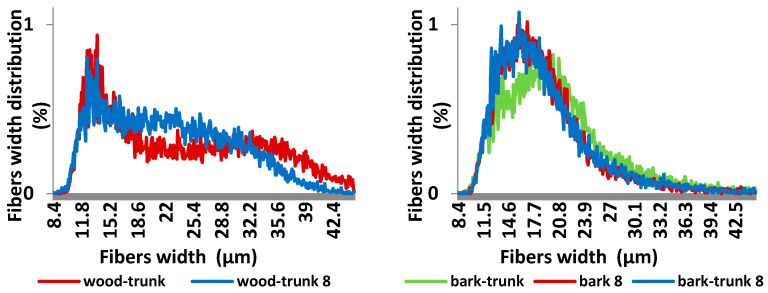
Fibres width distribution of spruce wood trunk and bark before and after storage for 8 months.

**Table 1 polymers-13-01619-t001:** Samples signification.

wood 0.5 m; 1.5 m; 2.5 m; 3.5 m; 4.5 m	sample taken from the tree trunk in a height of 0.5 m; 1.5 m; 2.5 m; 3.5 m; 4.5 m from the ground
wood-top	sample taken from the tree top
bark-trunk	sample taken from the trunk
bark-top	sample taken from the top
wood 0.5 m-A; 4.5 m-A	sample taken from the tree trunk in a height of 0.5 m or 4.5 m from the ground, first 20-years of growth (Figure 1)
wood 0.5 m-B; 4.5 m-B	sample taken from the tree trunk in a height of 0.5 m or 4.5 m from the ground, from 20 to 40 years of growth (Figure 1)
wood 0.5 m-C; 4.5 m-C	sample taken from the tree trunk in a height of 0.5 m or 4.5 m from the ground, the last 25-years of growth (Figure 1)
wood 0, 2, 4, 6, 8 month	wood from the trunk taken in time of 0, 2, 4, 6, and 8 month of its storage
bark trunk 0, 2, 4, 6, 8 month	bark from the trunk taken in time of 0, 2, 4, 6, and 8 month of its storage
bark 0, 2, 4, 6, 8 month	separately stored bark taken in time of 0, 2, 4, 6, and 8 month of its storage

**Table 2 polymers-13-01619-t002:** Conditions during wood storage.

Time of Storage/ Conditions	Average Air Humidity (%)	Average Air Temperature (°C)	Average Precipitation (mm)
from Jun to August	82.99 (max. 99.05, min. 59.32)	19.31 (max. 23.08, min. 14.37)	3.20 (max. 29.0, min. 0)
from August to October	86.89 (max. 99.17, min. 71.39)	17.02 (max. 23.63, min. 8.95)	2.28 (max. 23.80, min. 0)
from October to December	96.98 (max. 99.17, min. 81.52)	5.39 (max. 12.45, min. −2.99)	2.06 (max. 42.2, min. 0)
from December to February	97.45 (max. 99.17, min. 76.56)	0.54 (max. 6.17, min. −8.79)	1.50 (max. 17.8, min. 0)

**Table 3 polymers-13-01619-t003:** The average fibre length and width of wood samples.

Trait/Sample	0.5-A	0.5-B	0.5-C	4.5-A	4.5-B	4.5-C
average fiber length (mm)	0.83	1.33	1.55	1.22	1.61	1.65
average fiber width (µm)	21.58	23.58	23.91	23.26	24.49	24.70

**Table 4 polymers-13-01619-t004:** The average fibre length and width before and after storage for 8 months.

Trait/Sample	Wood-Trunk	Wood-Trunk 8	Bark-Trunk	Bark 8	Bark-Trunk 8
average fiber length (mm)	1.50	1.02	0.49	0.35	0.38
average fiber width (µm)	24.33	22.50	21.02	19.57	19.36

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
