# Peer review of "Chemical and Morphological Composition of Norway Spruce Wood (Picea abies, L.) in the Dependence of Its Storage"

_polymers, 2021, doi:10.3390/polym13101619_

Round 1

Reviewer 1 Report

Table 2 I recommend to state the minimum and maximum value of the parameter

or add to the supplement: graphs displaying temperature vs. storage periods, Humidity vs. storage period, precipitation vs. storage period

page 3 line 96: incorrect name of reference Wiese (correct Wise)

Use the term height (remove / replace high)

Has the ash content been measured? Fill in the data.

Can you state what the loss of mass after storage is?

From the literature, you can indicate which extractive substances / types predominate in the wood / bark after storage.

What causes changes to the wood biomass during storage? What mechanisms cause it, add information.

Add information on the main reactions that occur when storing wood in terms of extractives.

example: Routa, J., Brännström, H., Hellström, J. et al. Influence of storage on the physical and chemical properties of Scots pine bark. Bioenerg. Res. (2020). https://doi.org/10.1007/s12155-020-10206-8

Author Response

Response to Reviewer 1 Comments

Dear reviewer, thank you for your suggestions.

Table 2 I recommend to state the minimum and maximum value of the parameter or add to the supplement: graphs displaying temperature vs. storage periods, Humidity vs. storage period, precipitation vs. storage period

Answer: I stated the minimum and maximum value (Table 2)

page 3 line 96: incorrect name of reference Wiese (correct Wise)

Answer: I corrected it

Use the term height (remove / replace high)

Answer: I corrected it

Has the ash content been measured? Fill in the data.

Answer: We don´t measured the ash content

Can you state what the loss of mass after storage is?

Answer: We didn´t follow the loss of mass after storage.

From the literature, you can indicate which extractive substances / types predominate in the wood / bark after storage.

Answer: I added it

What causes changes to the wood biomass during storage? What mechanisms cause it, add information.

Answer: I added it

Add information on the main reactions that occur when storing wood in terms of extractives.

Answer: I added it

example: Routa, J., Brännström, H., Hellström, J. et al. Influence of storage on the physical and chemical properties of Scots pine bark. Bioenerg. Res. (2020). https://doi.org/10.1007/s12155-020-10206-8

Reviewer 2 Report

The manuscript entitled “Chemical and Morphological Composition of Norway Spruce 2 Wood (Picea abies, L.) in Dependence of Its Storage This work is merit for publication at Polymers after some major modification. So I have some points that may help to improve the work as follows:

1-Abstract is good but need more explain about the main aim of work

2- The introduction should be extended to discuss the hypothesis and research questions in details. Additionally, the introduction should cover the recent literature related to this subject.

3- Material and methods

The methodologies should be explained in details so that the results are reproducible.

4-Results

The results are clear and important.

5-Discussion
The discussion section still needs improvement, and should be linked to the findings of the previous reports on this topic.

6- The conclusion

A section for conclusions need more explain and should include the most significant findings and future works only.

7- English writing should be checked by a native English speaking expert.

Author Response

1-Abstract is good but need more explain about the main aim of work

Answer: We have no space for other explanation because of restriction of words number in abstract.

2- The introduction should be extended to discuss the hypothesis and research questions in details. Additionally, the introduction should cover the recent literature related to this subject.

Answer: I added the recent literature to the Introduction part.

3- Material and methods

The methodologies should be explained in details so that the results are reproducible.

Answer: Thank you for the opinion, this Methods was published many time as it is. There are presented standards and chemical analysis can be provided according them.

4-Results

The results are clear and important.

Answer: Thank you so much!

5-Discussion

The discussion section still needs improvement, and should be linked to the findings of the previous reports on this topic.

Answer: We added some discussion to this part.

6- The conclusion

A section for conclusions need more explain and should include the most significant findings and future works only.

Answer: We added some information.

7- English writing should be checked by a native English speaking expert.

Answer: we did so

Round 2

Reviewer 2 Report

The authors have made changes to the manuscript, so I consider it can be accepted for publication.